# LEARNING INTERACTIVE REAL-WORLD SIMULATORS

**Sherry Yang,**[1,2]   **Yilun Du**[3]   **Kamyar Ghasemipour**[2]   **Jonathan Tompson**[2]
**Leslie Kaelbling**[3]   **Dale Schuurmans**[2,4]   **Pieter Abbeel**[1]
[1]UC Berkeley   [2]Google DeepMind   [3]MIT   [4]University of Alberta
`sherryy@{berkeley.edu, google.com}`

## ABSTRACT

Generative models trained on internet data have revolutionized how text, image, and video content can be created. Perhaps the next milestone for generative models is to simulate realistic experience in response to actions taken by humans, robots, and other interactive agents. Applications of a real-world simulator range from controllable content creation in games and movies, to training embodied agents purely in simulation that can be directly deployed in the real world. We explore the possibility of learning a universal simulator (UniSim) of real-world interaction through generative modeling. We first make the important observation that natural datasets available for learning a real-world simulator are often rich along different dimensions (e.g., abundant objects in image data, densely sampled actions in robotics data, and diverse movements in navigation data). With careful orchestration of diverse datasets, each providing a different aspect of the overall experience, we can simulate the visual outcome of both high-level instructions such as "open the drawer" and low-level controls such as "move by $\Delta x, \Delta y$" from otherwise static scenes and objects. We use the simulator to train both high-level vision-language policies and low-level reinforcement learning policies, each of which can be deployed in the real world in zero shot after training purely in simulation. We also show that other types of intelligence such as video captioning models can benefit from training with simulated experience, opening up even wider applications. Video demos can be found at https://universal-simulator.github.io.

## 1 INTRODUCTION

Generative models trained on internet data can now produce highly realistic text (OpenAI, 2023),image (Ramesh et al., 2022), and video (Ho et al., 2022a). Perhaps the ultimate goal of generative models is to be able to simulate the visual effects of a wide variety of actions, from how cars are driven on a street to how furniture and meals are prepared. With a real-world simulator, humans can "interact" with diverse scenes and objects, robots can learn from simulated experience without risking physical damage, and a vast amount of "real-world" like data can be simulated to train other types of machine intelligence.

One roadblock to building this simulator lies in the datasets — different datasets cover different information that have to be brought together to simulate realistic experience. For instance, paired text-image data from the internet contains rich scenes and objects but little movement (Schuhmann et al., 2022; Zhai et al., 2022), video captioning and question answering data contain rich high-level descriptions but little low-level movement detail (Xu et al., 2016; Krishna et al., 2017), human activity data contains rich human action but little mechanical motion (Miech et al., 2019; Grauman et al., 2022), and robotics data contains rich robot action but are limited in quantity (Dasari et al., 2019; Mandlekar et al., 2018). Since different datasets are curated by different industrial or research communities for different purposes, divergence in information is natural and hard to overcome, posing difficulties to a real-world simulator that seeks to capture all visual aspects of the world.

In this work, we propose to combine a wealth of data in a conditional video generation framework to instantiate a universal simulator (UniSim)[1]. Under a unified action-in-video-out interface, the simulator enables rich interaction through fine-grained motion control of otherwise static scenes and objects. To support long-horizon repeated interactions, we formulate the simulator as an *observation*

---

[1]Note that by "universal", we mean the model can simulate through the unified interface of actions and videos, as opposed to being able to simulate everything. Sound, for instance, is not being simulated.

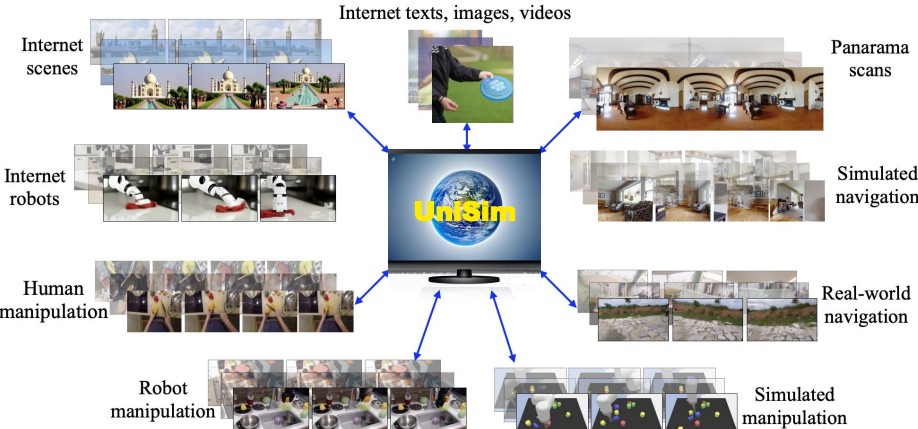

Figure 1: **A universal simulator (UniSim).** The simulator of the real-world learns from broad data with diverse information including objects, scenes, human activities, motions in navigation and manipulation, panorama scans, and simulations and renderings.

*prediction model* that can be rolled out autoregressively to support consistent simulation across video generation boundaries.

While the potential applications of the simulator are broad, we demonstrate three specific use cases. We first show how the simulator enables a vision-language policy to perform long-horizon goal-conditioned tasks through hindsight relabeling of simulated experience (Andrychowicz et al., 2017). In addition to learning high-level vision-language policies, we illustrate how the simulator can enable learning low-level control policies by leveraging model-based reinforcement learning (RL) (Sutton, 1988). Both the high-level vision-language policy and the low-level control policy, while trained purely in simulation, can generalize to real robot settings. This is enabled by using the simulator that is nearly visually indistinguishable from the real world, achieving one step towards bridging the sim-to-real gap in embodied learning (Rusu et al., 2017). Furthermore, we can simulate rare events where data collection is expensive or dangerous (e.g., crashes in self-driving cars). Such simulated videos can then be used to improve other machine intelligence such as rare event detectors, suggesting broad applications of UniSim beyond embodied learning. The main contributions can be summarized as follows:

- We take the first step toward building a universal simulator of real-world interaction by combining diverse datasets rich in along different dimensions — e.g., objects, scenes, actions, motions, language, and motor controls — in a unified action-in-video-out generative framework.
- We formulate the action-in-video-out framework as an *observation prediction model* conditioned on finite history and parametrized by a video diffusion model. We illustrate that the observation prediction model can be rolled out autoregressively to obtain consistent and long-horizon videos.
- We illustrate how the simulator can enable both high-level language policies, low-level control policies, and video captioning models to generalize to the real world when trained purely in simulation, thereby bridging the sim-to-real gap.

## 2 LEARNING AN INTERACTIVE REAL-WORLD SIMULATOR

We define a simulator of the real world as a model that, given some state of the world (e.g., an image frame), can take in some action as input, and produce the visual consequence of the action (in the form of a video) as output. Learning such a simulator is hard, since different actions have different formats (e.g.,language instructions, robot controls, camera movements) and videos have different frame rates. Nevertheless, we propose specific strategies for processing each type of data to unify the action space and align videos of variable lengths to actions in Section 2.1. With a unified action space, we then train an action-conditioned video generation model to fuse information across datasets through a universal interface relating actions to videos in Section 2.2.

### 2.1 ORCHESTRATING DIVERSE DATASETS

Below, we highlight diverse information in different datasets and propose ways to process actions into a common format (see all datasets used to train UniSim in Appendix B).

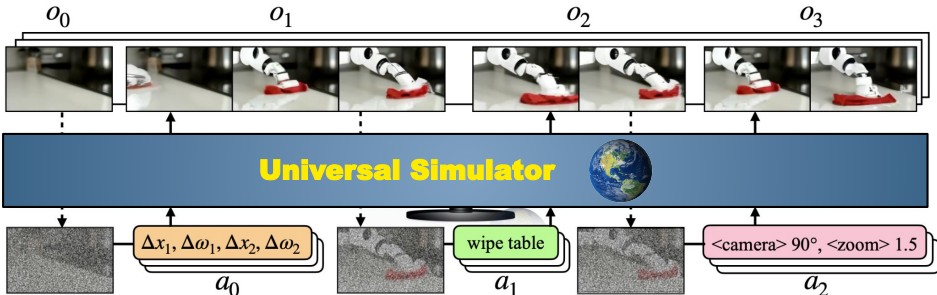

Figure 2: **Training and inference of UniSim.** UniSim is a video diffusion model trained to predict the next (variable length) set of observation frames ($o_t$) given observations from the past (e.g., $o_{t-1}$) and action input $a_{t-1}$. UniSim can handle temporally extended actions in various modalities such as motor controls ($\Delta x_1, \Delta \omega_1, \Delta x_2, ...$), language descriptions ("wipe table"), and actions extracted from camera motions and other sources. Each dotted arrow indicates concatenating the initial noise sample for the next video segment with the previous frame.

- **Simulated execution and renderings.** While annotating actions for real-world videos is expensive, simulation engines such as Habitat (Savva et al., 2019) can render a wide variety of actions. We use datasets previously collected from these simulators, i.e., Habitat object navigation with HM3D (Ramakrishnan et al., 2021) and Language Table Data from Lynch et al. (2023) to train UniSim. We extract text descriptions as actions when available. For simulated continuous control actions, we encode them via language embeddings and concatenate the text embeddings with discretized control values.
- **Real robot data.** An increasing amount of video data of real-robot executions paired with task descriptions such as the Bridge Data (Ebert et al., 2021) and data that enabled RT-1 and RT-2 (Brohan et al., 2022) are becoming increasingly available. Despite low-level control actions often being different across robots, the task descriptions can serve as high-level actions in UniSim. We further include discretize continuous controls actions when available similar to simulated robotics data.
- **Human activity videos.** Rich human activity data such as Ego4D (Grauman et al., 2022), EPIC-KITCHENS (Damen et al., 2018), and Something-Something V2 (Goyal et al., 2017) have been curated. Different from low-level robot controls, these activities are high-level actions that humans take to interact with the world. But these actions are often provided as labels for video classification or activity recognition tasks (Goyal et al., 2017). In this case, we convert the video labels into text actions. In addition, we subsample the videos to construct chunks of observations at a frame rate that captures meaningful actions.
- **Panorama scans.** There exists a wealth of 3D scans such as Matterport3D (Chang et al., 2017). These static scans do not contain actions. We construct actions (e.g., turn left) by truncating panorama scans and utilize the information of camera poses between two images.
- **Internet text-image data.** Paired text-image datasets such as LAION (Schuhmann et al., 2021) contain static images of a variety of objects without actions. However, the captions often contain motion information such as "a person *walking*". To use image data in UniSim, we treat individual images as single-frame videos and image captions as actions.

For each of these datasets, we process text tokens into continuous representations using T5 language model embeddings (Raffel et al., 2020) concatenated with low-level actions such as robot controls. This serves as the final unified action space of our simulator.

## 2.2 SIMULATING LONG-HORIZON INTERACTIONS THROUGH OBSERVATION PREDICTION

With observations from different environments that have been converted to videos, and actions of different formats that have been converted to continuous embeddings, we can formulate interactions with many real-world environments as interacting with a universal simulator. We then formulate the universal simulator as an *observation prediction model* that predicts observations conditioned on actions and previous observations as shown in Figure 2. We finally show that this observation prediction model can be parametrized using video diffusion.

**Simulating Real-World Interactions.** We define an observation space $O$ and an action space $A$ which capture the videos and actions described in Section 2.1. At a specific interactive step $t$, an agent, having observed a set of history frames $h_{t-1} \in O$, decides on some temporally extended action $a_{t-1} \in A$, which can be resolved into a sequence of low-level robot commands to be executed in

the real world. During the execution, the next set of video frames $o_t \in O$ are captured from the real world. The goal of a simulator is to predict $o_t$ from $h_{t-1}$ and $a_{t-1}$. We can formulate this prediction problem as learning an *observation prediction* model $p(o_t|h_{t-1}, a_{t-1})$. While an ideal predictive model should condition on all information of the past, i.e., $(o_0, a_0 \ldots, a_{t-2}, o_{t-1})$, through some recurrent state, we found conditioning on a finite set of frames (e.g., frames from the most recent interaction, $o_{t-1}$) greatly simplifies the modeling problem. To simulate long interactions, we can sample from the observation prediction model $p(o_t|h_{t-1}, a_{t-1})$ autoregressively conditioned on the previously sampled observations. One advantage of this observation prediction model is that the simulator stays the same across all tasks and can be used in combination with any reward function, which can be separately learned. The learned reward function can then be used to optimize policies $\pi(a_t|h_t)$ using existing decision making algorithms such as planning and RL, as we will illustrate in Section 4.1 and Section 4.2.

**Parametrizing and Training the Simulator.** We parametrize $p(o_t|h_{t-1}, a_{t-1})$ using diffusion models (Sohl-Dickstein et al., 2015; Ho et al., 2020) as an instantiation of UniSim outlined in Figure 2. Specifically, the reverse process learns a denoising model $\epsilon_\theta(o_t^{(k)}, k|h_{t-1}, a_{t-1})$ that, conditioned on the history, generates the next observationfrom initial noise samples using $K$ denoising steps. In practice, we only use previous video frames and omit previous actions as history, and concatenate previous video frames with initial noise samples $o_t^{(K)} \sim \mathcal{N}(0, I)$ channelwise to serve as conditional inputs to the denoising model. To condition on an action $a_{t-1}$, we leverage classifier-free guidance (Ho & Salimans, 2022). The final $\overline{\mathcal{T}}(o_t|h_{t-1}, a_{t-1})$ is parametrized by the variance schedule:

$$\epsilon_\theta(o_t^{(k)}, k|h_{t-1}, a_{t-1}) = (1 + \eta)\epsilon_\theta(o_t^{(k)}, k|h_{t-1}, a_{t-1}) - \eta\epsilon_\theta(o_t, k|h_{t-1}), \qquad (1)$$

where $\eta$ controls action conditioning strength. With this parametrization, we train $\epsilon_\theta$ by minimizing

$$\mathcal{L}_{\text{MSE}} = \left\| \epsilon - \epsilon_\theta\left(\sqrt{1 - \beta^{(k)}}o_t + \sqrt{\beta^{(k)}}\epsilon, \, k \Big| h_{t-1}, a_{t-1}\right) \right\|^2,$$

where $\epsilon \sim \mathcal{N}(0, I)$, and $\beta^{(k)} \in \mathbb{R}$ are a set of $K$ different noise levels for each $k \in [1, K]$. Given the learned $\epsilon_\theta$, an observation $o_t$ can be generated by sampling from the initial distribution $o_t^{(K)} \sim \mathcal{N}(0, I)$ and iteratively denoising according to the following process for $k$ from $K$ to 0

$$o_t^{(k-1)} = \alpha^{(k)}(o_t^{(k)} - \gamma^{(k)}\epsilon_\theta(o_t^{(k)}, k|h_{t-1}, a_{t-1})) + \xi, \quad \xi \sim \mathcal{N}(0, \sigma_k^2 I), \qquad (2)$$

where $\gamma^{(k)}$ is the denoising step size, $\alpha^{(k)}$ is a linear decay on the current denoised sample, and $\sigma_k$ is a time varying noise level that depends on $\alpha^{(k)}$ and $\beta^{(k)}$.

**Architecture and Training.** We use the video U-Net architecture (Ho et al., 2022b) to implement UniSim by employing interleaved temporal and spatial attention and convolution layers in both the downsampling and upsampling passes. For history conditioning, we replicate the conditioning frames at all future frame indices, and concatenate the conditioning frames with the noise sample for each of the future frame to serve as input to the U-Net. UniSim model has 5.6B parameters and requires 512 TPU-v3 and 20 days to train on all data. See more details in Appendix C.

## 3 SIMULATING REAL-WORLD INTERACTIONS

We now demonstrate emulating real-world manipulation and navigation environments by simulating both action-rich and long-horizon interactions for both humans and robots.

### 3.1 ACTION-RICH, LONG-HORIZON, AND DIVERSE INTERACTIONS

**Action-Rich Simulation.** We first demonstrate action-rich interactions through natural language actions. Figure 3 shows simulation of human manipulation and navigation starting from the same initial observation (left-most column). We can instruct a person in the initial frame to perform various kitchen tasks (top left), press different switches (top right), or navigate scenes (bottom). The model only trained on generic internet data, without action-rich manipulation data such as EPIC-KITCHENS (Damen et al., 2018), fails to simulate action-rich manipulations (Appendix F).

**Long-Horizon Simulation.** Next, we illustrate 8 *sequential* interactions in Figure 4. We condition the simulation of each interaction on previous observations and new language action as described in Section 2.2. UniSim successfully preserves objects manipulated by previous instructions (e.g., the orange and can are preserved in the drawers in Columns 4, 5, 7, 8 after being put in the drawers). See additional long-horizon interactions in Appendix A.1.

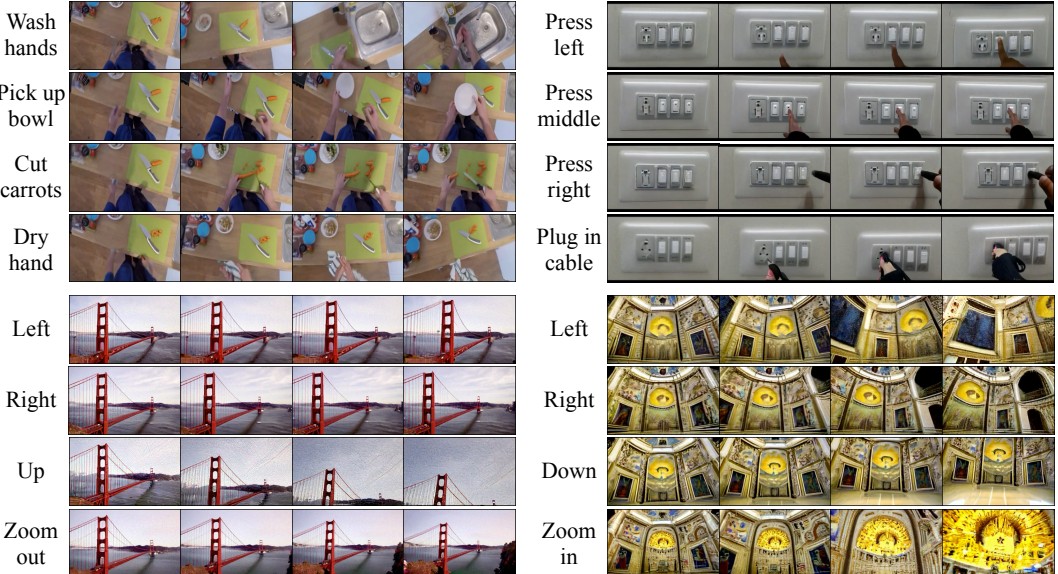

Figure 3: **Action-rich simulations.** UniSim can support manipulation actions such as "cut carrots", "wash hands", and "pickup bowl" from the same initial frame (top left) and other navigation actions.

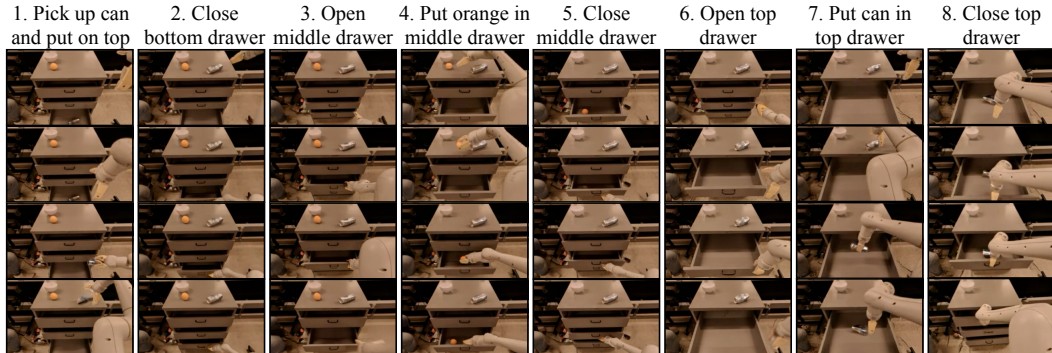

Figure 4: **Long-horizon simulations.** UniSim sequentially simulates 8 interactions autoregressively. The simulated interactions maintain temporal consistency across long-horizon interactions, correctly preserving objects and locations (can on counter in column 2-7, orange in drawer in column 4-5).

| Condition | FID ↓ | FVD ↓ | IS ↑ | CLIP ↑ |
|---|---|---|---|---|
| 1 frame | 59.47 | 315.69 | 3.03 | 22.55 |
| 4 distant | 34.89 | 237 | 3.43 | 22.62 |
| 4 recent | **34.63** | **211.3** | **3.52** | **22.63** |

Table 1: **Ablations of history conditioning** using FVD, FID, and Inception score, and CLIP score on Ego4D. Conditioning on multiple frames is better than on a single frame, and recent history has an edge over distant history.

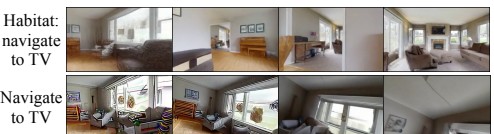

Figure 6: **Simulations of low-data domains** using the Habitat object navigation using HM3D dataset (Ramakrishnan et al., 2021) with only 700 training examples. Prefixing language actions with dataset identifier leads to video samples that complete the action (top).

**Diversity and Stochasticity in the Simulator.** UniSim can also support highly diverse and stochastic environment transitions, e.g., diverse objects being revealed after removing the towel on top (Figure 5 left), diverse object colors and locations (cups and pens in Figure 5 right), and real-world variabilities such as change in camera angles. Flexibility in diffusion models promotes simulation of highly stochastic environments that cannot be controlled by actions, so that a policy can learn to only control the controllable part (Yang et al., 2022).

## 3.2 ABLATION AND ANALYSIS

**Frame Conditioning Ablations.** We ablate over choices of past frames to condition on using a validation split of the Ego4D dataset (Grauman et al., 2022), which contains egocentric movement requiring proper handling of observation history. We compare UniSim conditioned on different

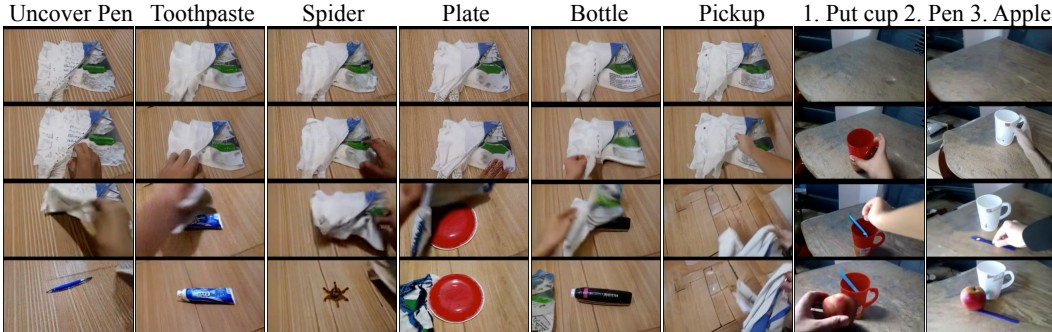

| Uncover Pen | Toothpaste | Spider | Plate | Bottle | Pickup | 1. Put cup 2. Pen 3. Apple |

Figure 5: **Diverse and stochastic simulations.** On the left, we use text to specify the object being revealed by suffixing "uncovering" with the object name. On the right, we only specify "put cup" or "put pen", and cups and pens of different colors are sampled as a result of the stochastic sampling process during video generation.

numbers of past frames in Table 1. Conditioning on 4 frames is better than conditioning on a single frame, but conditioning on history that is too far in the past (4 frames with exponentially increasing distances) can hurt performance. Increasing the number of conditioning frames beyond 4 did not further improve performance on Ego4D, but it could be helpful for applications that require memory from distant past (e.g., navigation for retrieval).

**Simulating Low-Data Domains.** During joint training of UniSim on diverse data, we found that naïvely combining datasets of highly varying size can result in low generation quality in low-data domains. While we can increase the weight of these domains in the data mixture during training, we found that attaching a domain identifier such as the name of the dataset to the actions being conditioned on improves generation quality in low-data domains, as shown in Figure 6. While such domain identifier improves in-distribution generation quality, we found domain-specific identifiers to hurt generalization to other domains, and should only be applied with the test domain is in distribution of the training domain.

## 4 APPLICATIONS OF UNISIM

We now demonstrate how UniSim can be used to train other types of machine intelligence such as vision-language policies, RL agents, and vision-language models through simulating highly realistic experiences.

### 4.1 TRAINING LONG-HORIZON VISION-LANGUAGE POLICIES THROUGH HINDSIGHT LABELING.

Language models and vision language models (VLM) have recently been used as policies that can operate in image or text based observation and action spaces (Du et al., 2023b; Driess et al., 2023; Brohan et al., 2023). One major challenge in learning such agents lies in the need for large amounts of language action labels. The labor intensity in data collection only increases as tasks increase in horizon and complexity. UniSim can generate large amounts of training data for VLM policies through hindsight relabeling.

**Setup and Baseline.** We use data from the Language Table environment (Lynch & Sermanet, 2020) for learning geometric rearrangements of blocks on a table. We train an image-goal conditioned VLM policy to predict language instructions and the motor controls from the start and goal images using the PALM-E architecture (Driess et al., 2023) (See data and model details in Appendix D.1). For the baseline, the goal is set to the last frame of the original short-horizon trajectories. During each evaluation run, we set the long-horizon goal by modifying the location of 3-4 blocks, and measure the blocks' distance to their goal states after executing 5 instructions using the VLM policy. We define the reduction in distance to goal (RDG) metric as

$$\text{RDG} = \frac{\|s_0 - s_{\text{goal}}\|_2 - \|s_T - s_{\text{goal}}\|_2}{\|s_0 - s_{\text{goal}}\|_2}, \tag{3}$$

where $s_T$ represents the underlying block locations after executing the policy, $s_0$ and $s_{\text{goal}}$ represents the initial and goal block locations.

**Generating Hindsight Data with the Simulator.** To use the simulator for long-horizon tasks, we draw inspiration from hindsight relabeling (Rauber et al., 2019). Specifically, we create a total of 10k long-horizon trajectories from the simulator by doing rollouts in the simulator 3-5 times per trajectory, where each rollout corresponds to one scripted language instruction. We then use the

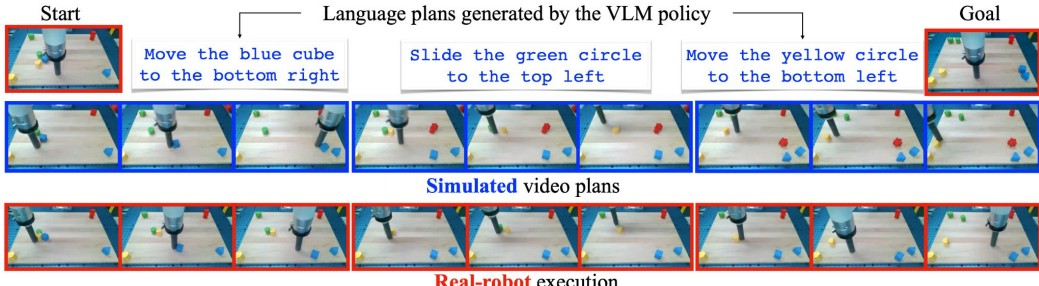

Figure 7: **Long-horizon simulation.** A VLM poliy generates high-level language actions (first row) which are executed in the simulator (middle row) similar to how they are executed in the real world (bottom row) using the Language Table robot. The VLM trained on data from the simulator complete long-horizon tasks by successfully moving three blocks (blue, green, yellow) to match their target location in the goal image.

|  | RDG (moved) | RDG (all) |
|---|---|---|
| VLM-BC | $0.11 \pm 0.13$ | $0.07 \pm 0.11$ |
| Simulator-Hindsight | $\mathbf{0.34} \pm 0.13$ | $\mathbf{0.34} \pm 0.13$ |

|  | Succ. rate (all) | Succ. rate (pointing) |
|---|---|---|
| VLA-BC | 0.58 | 0.12 |
| Simulator-RL | **0.81** | **0.71** |

Table 2: **Evaluation of long-horizon actions.** Reduction in distance to goal (RDG) defined in Equation 3 across 5 evaluation runs of VLM trained using simulated long-horizon data (bottom row) compared to VLM trained on original short-horizon data (top row). Using the simulator performs much better both in RGD of moved blocks (left) and RGD in all blocks (right).

Table 3: **Evaluation of RL policy.** Percentage of successful simulated rollouts (out of 48 tasks) using the VLA policy with and without RL finetuning on Language Table (assessed qualitatively using video rollouts in the simulator). Simulator-RL improves the overall performance, especially in pointing-based tasks which contain limited expert demonstrations.

final frame from each long-horizon rollout as a goal input and the scripted language instructions as supervision for training the VLM policy.

**Results on Real-Robot Evaluation.** Despite the VLM policy only being trained on simulated data, it is able to produce effective high-level language actions given an initial and goal image from the real Language Table domain where the data for training the simulator was collected. The simulator can simulate video trajectories from the initial real observation, from which robot actions are recovered using an inverse dynamics model and executed on the real robot. Figure 7 shows that the language actions produced by the VLM, the generated videos from the simulator according to the language actions, and the executions on the real robot. We see that the simulated video trajectory is successfully translated to robot actions in the real world. See additional results from the long-horizon VLM policy in Appendix A.2.

**Results on Simulated Evaluation.** In addition to testing the language instructions and simulated video by converting video trajectory into robot actions executed on the real robot, we also conduct simulator based evaluation to compare the reduction in distance to goal (RDG) of the VLM policy using generated long-horizon data to using the original short-horizon data in Table 2. The VLM trained using long-horizon generated data performs 3-4 times better than using the original data in completing long-horizon goal-conditioned tasks.

## 4.2 REAL-WORLD SIMULATOR FOR REINFORCEMENT LEARNING

Reinforcement learning (RL) has achieved superhuman performance on difficult tasks such as playing Go and Atari games (Silver et al.; Mnih et al., 2015), but has limited real world applications due, among other reasons, to the lack of a realistic environment simulator (Dulac-Arnold et al., 2019). We investigate whether the simulator can enable effective training of RL agents by providing the agent with a realistic simulator that can be accessed in parallel.

**Setup.** We finetune the PaLI 3B vision-language model (Chen et al., 2022b) to predict low-level control actions (joint movements in $\Delta x, \Delta y$ from an image observation and a task description (e.g., "move the blue cube to the right") using behavioral cloning (BC) to serve as the low-level control policy and the baseline, which we call the vision-language-action (VLA) policy similar to Brohan et al. (2023). Because UniSim can take low-level control actions as input, we can directly conduct model-based rollouts in the simulator using control actions generated by VLA policy. To acquire reward information, we use the number of steps-to-completion from the training data as a proxy reward to train a model that maps the current observation to learned reward. We then use the

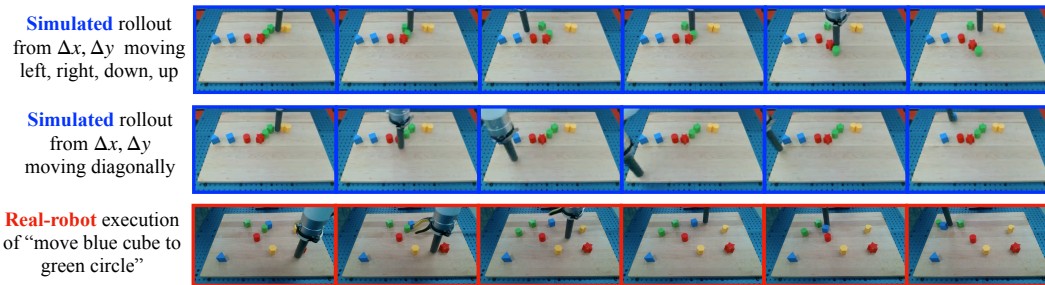

Figure 8: **[Top] Simulation from low-level controls**. UniSim supports low-level control actions as inputs to move endpoint horizontally, vertically, and diagonally. **[Bottom] Real-robot execution of an RL policy** trained in simulation and zero-shot onto the real Language Table task. The RL policy can successfully complete the task of "moving blue cube to green circle".

REINFORCE algorithm (Williams, 1992) to optimize the VLA policy, treating the rollouts from the simulator as the on-policy rollouts from the real environment and use the learned reward model to predict rewards from simulated rollouts. See details of RL training in Appendix D.2.

**Results.** We first do a sanity check on simulating real-robot executions by applying low-level control actions (e.g., $\Delta x = 0.05, \delta y = 0.05$) repeatedly for 20-30 environment steps to move the endpoint left, right, down, up, and diagonally in Figure 8 (top two rows). We see that the simulated rollouts capture both the endpoint movements and the physics of collision. To compare the RL policy trained in simulation to the BC policy, we qualitatively assessed the simulated rollouts in the simulator. Table 3 shows that RL training significantly improves the performance of the VLA policy across a wide set of tasks, especially in tasks such as "point to blue block". We then directly deploy the RL policy trained in the simulator onto the real robot in zero-shot, and observe successful task executions as shown in Figure 8 (bottom row). Additional results on real robot can be found in Appendix A.3.

### 4.3 REALISTIC SIMULATOR FOR BROADER VISION-LANGUAGE TASKS

UniSim can generate training data for other machine-learning subproblems. This is especially useful when natural data is rare or difficult to collect (e.g., footage of crimes or accidents). We provide such a proof-of-concept by training vision-language models on purely generated data from UniSim, and observe significant performance benefits in video captioning.

|  | Activity | MSR-VTT | VATEX | SMIT |
|---|---|---|---|---|
| No finetune | 15.2 | 21.91 | 13.31 | 9.22 |
| Activity | 54.90 | 24.88 | 36.01 | 16.91 |
| Simulator | 46.23 | **27.63** | **40.03** | **20.58** |

Table 4: **VLM trained in UniSim** to perform video captioning tasks. CIDEr scores for PaLI-X finetuned only on simulated data from UniSim compared to no finetuning and finetuning on true video data from ActivityNet Captions. Finetuning only on simulated data has a large advantage over no finetuning and transfers better to other tasks than finetuning on true data.

**Setup.** We finetune PaLI-X (Chen et al., 2023), a VLM with 55B parameters pretrained on a broad set of image, video, and language tasks, to caption a set of videos generated by UniSim using texts from the training split of ActivityNet Captions (Krishna et al., 2017). We measure the CIDEr score of the finetuned model on the test split of ActivityNet Captions as well as other captioning tasks following the same setup as Chen et al. (2023). See finetuning details of PaLI-X in Appendix D.3.

**Results.** We compare PaLI-X finetuned on purely generated videos to pretrained PaLI-X without finetuning and PaLI-X finetuned on original ActivityNet Captions in Table 4. Purely finetuning on generated data drastically improves the captioning performance from no finetuning at all on ActivityNet (15.2 to 46.23), while achieving 84% performance of finetuning on true data. Furthermore, PaLI-X finetuned on generated data transfers better to other captioning tasks such as MSR-VTT (Xu et al., 2016), VATEX (Wang et al., 2019), and SMIT (Monfort et al., 2021) than PaLI-X finetuned on true data, which tends to overfit to ActivityNet. These results suggest that UniSim can serve as an effective data generator for improving broader vision-language models.

### 5 RELATED WORK

**Internet-Scale Generative Models.** Language models trained on internet text succeed at text-based tasks (OpenAI, 2023; Anil et al., 2023) but not physical tasks, which requires perception and control. Internet-scale generative models can synthesize realistic images and videos (Wu et al., 2021; Ho et al., 2022a; Singer et al., 2022; Yang et al., 2023; Blattmann et al., 2023), but have mostly

been applied to generative media (Zhang et al., 2023) as opposed to empowering sophisticated agents capable of multi-turn interactions. Du et al. (2023a) shows video generation can serve as policies, but the major bottleneck for policy learning often lies in limited access to real-world environments (Dulac-Arnold et al., 2019). We focus on this exact bottleneck by learning universal simulators of the real world, enabling realistic and unlimited "environment" access for training sophisticated agents interactively.

**Learning World Models.** Learning an accurate dynamics model in reaction to control inputs has been a long-standing challenge in system identification (Ljung & Glad, 1994), model-based reinforcement learning (Sutton, 1991), and optimal control Åström & Wittenmark (1973); Bertsekas (1995). Most systems choose to learn one dynamics model per system in the lower dimensional state space as opposed to in the pixel space (Ferns et al., 2004; Achille & Soatto, 2018; Lesort et al., 2018; Castro, 2020), which, despite being a simpler modeling problem, limits knowledge sharing across systems. With large transformer architectures, learning image-based world models has become plausible (Hafner et al., 2020; Chen et al., 2022a; Seo et al., 2022; Micheli et al., 2022; Wu et al., 2022; Hafner et al., 2023), but mostly in games or simulated domains with visually simplistic and abundant data. In generative modeling of videos, previous works have leveraged text prompts (Yu et al., 2023; Zhou et al., 2022), driving motions (Siarohin et al., 2019; Wang et al., 2022), 3D geometries (Weng et al., 2019; Xue et al., 2018), physical simulations (Chuang et al., 2005), frequency information (Li et al., 2023), and user annotations (Hao et al., 2018) to introduce movements into videos. However, they focus on generating domain specific videos (e.g., for self-driving) as opposed to building a universal simulator that can be used to further improve other agents. The amount of control over generated videos in these existing work is also limited, as they do not treat video generation as a dynamics modeling problem like in our work.

## 6  LIMITATIONS AND CONCLUSION

We have shown it is possible to learn a simulator of the real world in response to various action inputs ranging from texts to robot controls. UniSim can simulate visually realistic experiences for interacting with humans and training autonomous agents. We hope UniSim will instigate broad interest in learning and applying real-world simulators to improve machine intelligence. Our simulator has a few limitations that call for future work:

- **Hallucination.** When an action is unrealistic given the scene (e.g., "wash hands" is given to a tabletop robot), we observe hallucinations (e.g., the table turns into a sink or the view turns away from the tabletop robot and a sink shows up). Ideally, we want UniSim to detect actions that are not possible to simulate as opposed to hallucinating unrealistic outcomes.
- **Limited memory.** The simulator conditioned on a few frames of the recent history cannot capture long-term memory (e.g., an apple in a drawer could disappear when the drawer is opened if putting the apple in the drawer is not a part of the history for conditioning). How much history to condition on depends on the application of the (e.g., whether the simulator will be used for policy learning in a near-Markov setting or question answering that requires long-term memory).
- **Limited out-of-domain generalization.** This is especially true for domains that are not represented in the training data. For instance, the simulator is mostly trained on 4 robot morphologies, and its ability to generalize to an unseen robot is limited. Further scaling up training data could help, as the training data is nowhere near all the video data available on the internet.
- **Visual simulation only.** Our simulator is not suitable for environments where actions do not cause visual observation change (e.g., different forces in grasping a static cup). A true universal simulator should capture all aspects of the world beyond visual experience (e.g., sound, sensory, etc).

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

# Appendix

In this Appendix we provide additional qualitative results on long-horizon simulation of human and robot interactions (Section A.1), long-horizon VLM policies (Section A.2), and low-level RL policies (Section A.3) that work on real robot. We also provided details on the dataset used to train UniSim in Section B, the model architecture and training details of UniSim in Section C, and the details of the three experimental setups for applications of UniSim in Section D. Finally, we provide failed examples when UniSim is not jointly trained on broad datasets (Section F). Video demos can be found at anonymous-papers-submissions.github.io

## A    ADDITIONAL RESULTS

### A.1    ADDITIONAL LONG-HORIZON INTERACTION

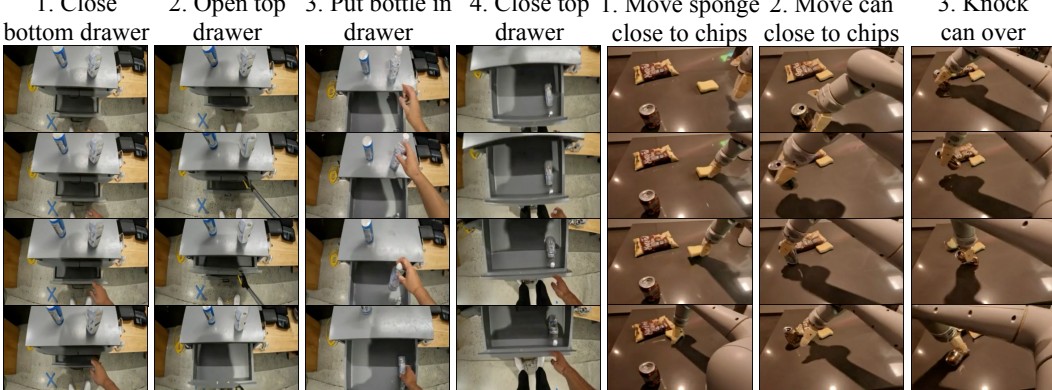

Figure 9: Additional results on long-horizon interaction with humans and robots similar to Figure 4. UniSim can generate consistent video rollouts across 3-4 high-level language actions.

## A.2 ADDITIONAL REAL-ROBOT RESULTS FOR LONG-HORIZON LANGUAGE POLICY

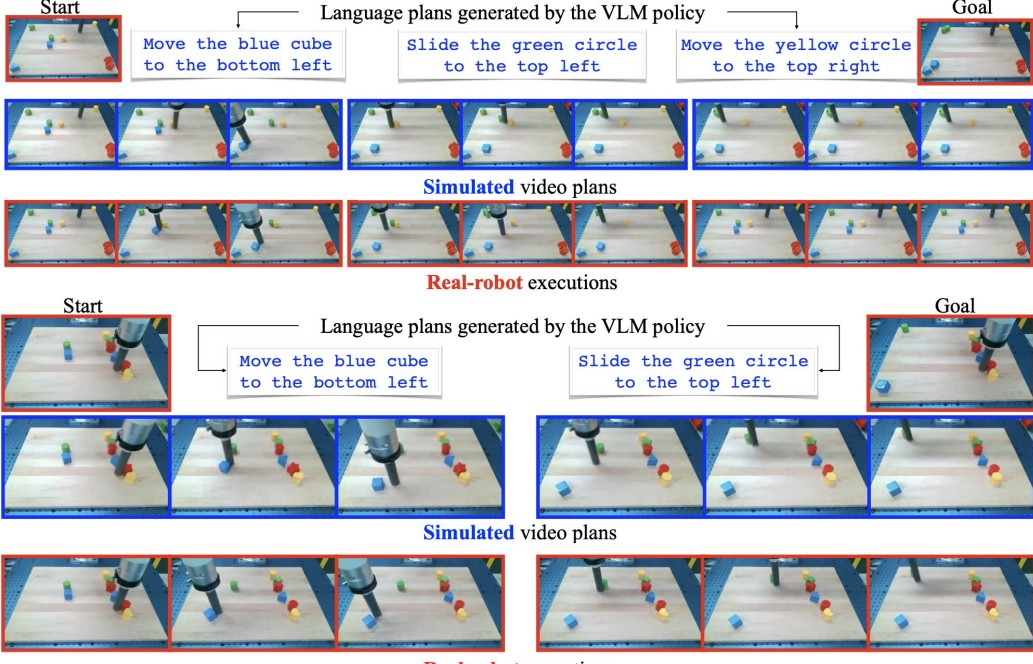

Figure 10: Additional results (similar to Figure 7) on applying UniSim to train vision-language policies to complete long-horizon tasks. VLM finetuned with hindsight labeled data is able to generate long-horizon instructions that moves two or three blocks successfully to match their location in the goal image.

## A.3 Additional Results on Learning RL Policy in UniSim

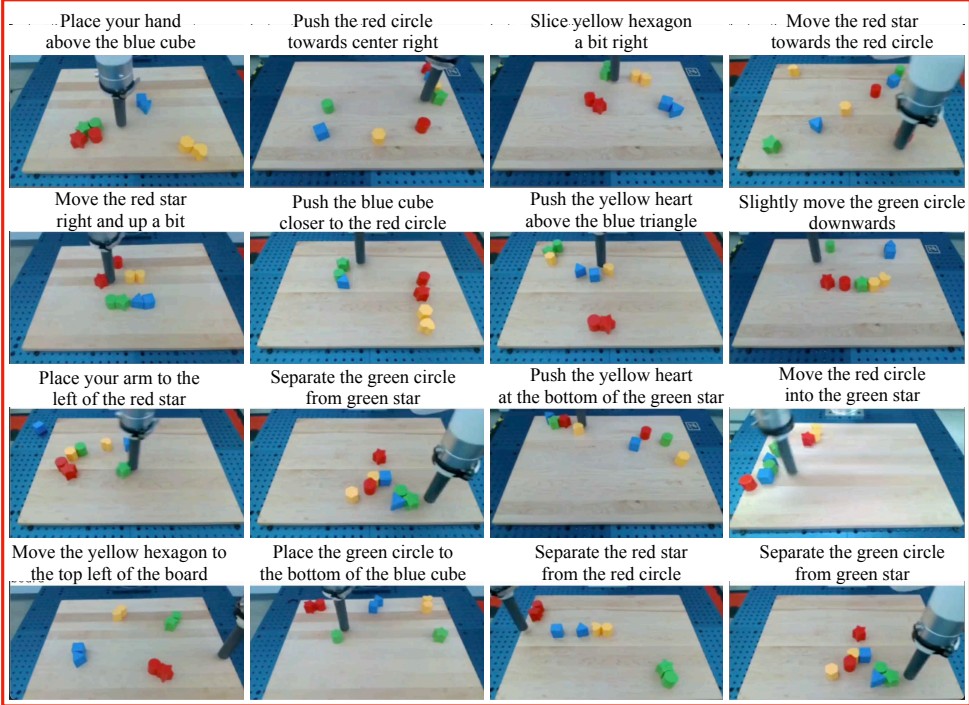

Figure 11: First real observations and last simulated observations of rolling out the RL policy trained in UniSim.

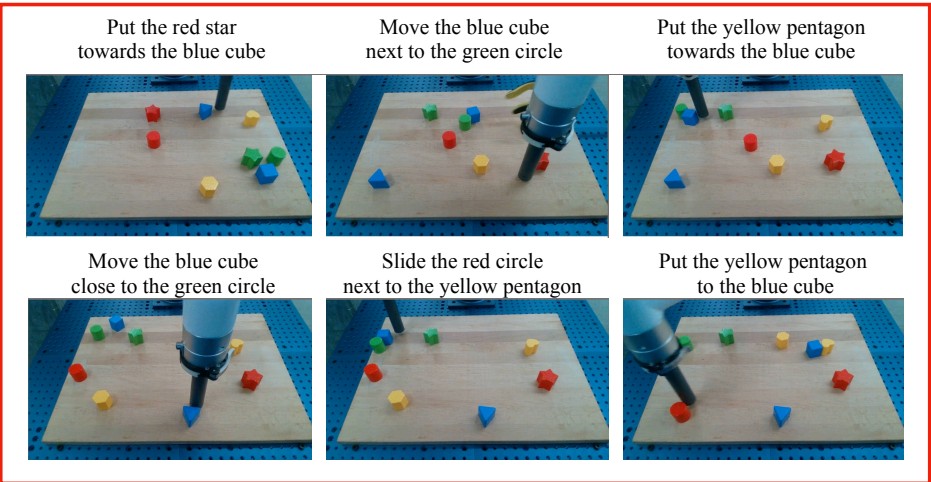

**Real first** observation of each trajectory

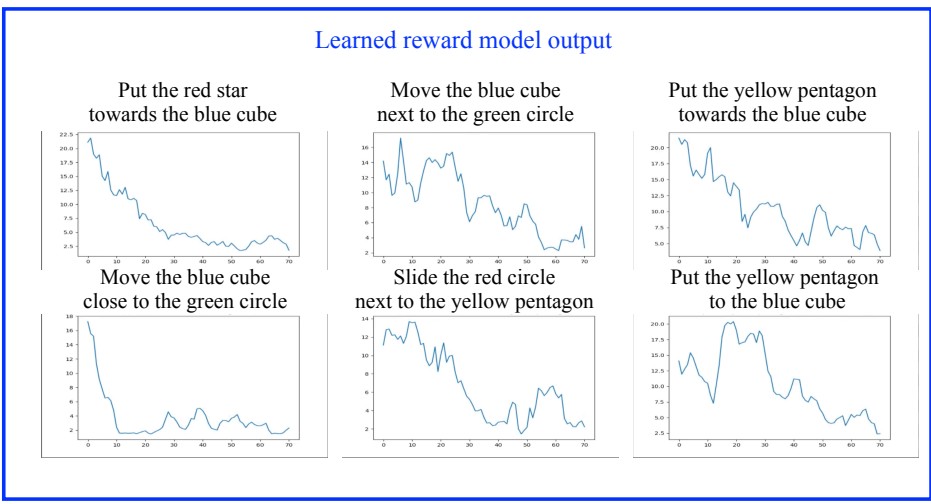

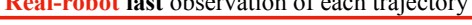

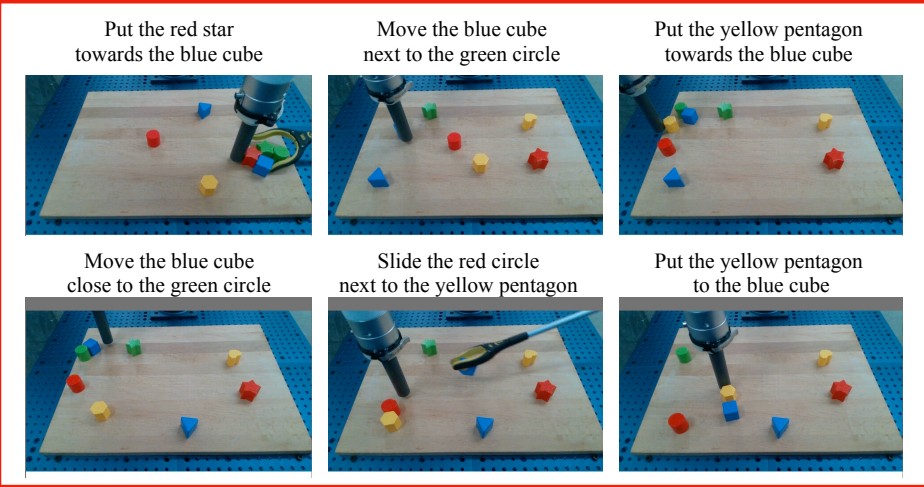

Figure 12: First real observations and last real observations of executing the RL policy trained from UniSim in the real world in zero-shot. Middle plot also shows the output of the learned reward model (steps-to-completion) during policy execution, where step 0 corresponds to the top plot (initial observation) and step 70 corresponds to the bottom plot (final observation).

# B   DATASETS

We provide the datasets used to train UniSim below, including dataset name, number of training examples (approximate), and weight in the data mixture. Miscellaneous data are collections of datasets that have not been published. Some of these datasets have been processed into train and validation split, hence the number of training examples may differ from the original data size. When text are available in the original dataset, we use T5 language model embeddings (Raffel et al., 2020) to preprocess the text into continuous representations. When low-level controls are available in the original dataset, we encode them both as text and normalize then discretize them into 4096 bins contatenated with language embeddings (if present). The choice of mixture weights are either 0.1 or 0.05 without careful tuning. How data mixture weights affect simulation performance is an interesting line of future work.

|  | Dataset | # Examples | Weight |
|---|---|---|---|
| Simulation | Habitat HM3D (Ramakrishnan et al., 2021) | 710 | 0.1 |
| | Language Table sim (Lynch & Sermanet, 2020) | 160k | 0.05 |
| Real Robot | Bridge Data (Ebert et al., 2021) | 2k | 0.05 |
| | RT-1 data (Brohan et al., 2022) | 70k | 0.1 |
| | Language Table real (Lynch & Sermanet, 2020) | 440k | 0.05 |
| | Miscellaneous robot videos | 133k | 0.05 |
| Human activities | Ego4D (Grauman et al., 2022) | 3.5M | 0.1 |
| | Something-Something V2 (Goyal et al., 2017) | 160k | 0.1 |
| | EPIC-KITCHENS (Damen et al., 2018) | 25k | 0.1 |
| | Miscellaneous human videos | 50k | 0.05 |
| Panorama scan | Matterport Room-to-Room scans (Anderson et al., 2018) | 3.5M | 0.1 |
| Internet text-image | LAION-400M (Schuhmann et al., 2021) | 400M | 0.05 |
| | ALIGN (Jia et al., 2021) | 400M | 0.05 |
| Internet video | Miscellaneous videos | 13M | 0.05 |

Table 5: Dataset name, number of training examples, and mixture weights used for training UniSim.

## C  ARCHITECTURE AND TRAINING

We the 3D U-Net architecture (Çiçek et al., 2016; Ho et al., 2022b) to parametrize UniSim video model. We apply the spatial downsampling pass followed by the spatial upsampling pass with skip connections to the downsampling pass activations with interleaved 3D convolution and attention layers as in the standard 3D U-Net. The video models in UniSim consist of one history conditioned video prediction model as the base and two additional spatial super-resolution models similar to Ho et al. (2022a). The history conditioned base model operates at temporal and spatial resolution $[16, 24, 40]$, and the two spatial super-resolution models operate at spatial resolution $[24, 40] \rightarrow [48, 80]$ and $[48, 80] \rightarrow [192, 320]$, respectively. To condition the base video model on the history, we take 4 frames from the previous video segment and concatenate them channelwise to the noise samples inputted to the U-Net. We employ temporal attention for the forward model to allow maximum modeling flexibility but temporal convolution to the super-resolution models for efficiency reasons similar to Ho et al. (2022a). The model and training hyperparamters of UniSim are summarized in Table 6.

| Hyperparameter | Value |
|---|---|
| Base channels | 1024 |
| Optimizer | Adam ($\beta_1 = 0.9, \beta_2 = 0.99$) |
| Channel multipliers | 1, 2, 4 |
| Learning rate | 0.0001 |
| Blocks per resolution | 3 |
| Batch size | 256 |
| Attention resolutions | 6, 12, 24 |
| Num attention heads | 16, 16, 8 |
| Conditioning embedding dimension | 4096 |
| Conditioning embedding MLP layers: 4 | |
| Conditioning token length | 64 |
| EMA | 0.9999 |
| Dropout | 0.1 |
| Training hardware | 512 TPU-v3 chips |
| Training steps | 1000000 |
| Diffusion noise schedule | cosine |
| Noise schedule log SNR range | [-20, 20] |
| Sampling timesteps | 256 |
| Sampling log-variance interpolation | $\gamma = 0.1$ |
| Weight decay | 0.0 |
| Prediction target | $\epsilon$ |

Table 6: Hyperparameters for training UniSim diffusion model.

# D DETAILS OF EXPERIMENTAL SETUPS

## D.1 DETAILS OF LEARNING LONG-HORIZON POLICY

**Language Table Dataset and environment.** The Language Table (Lynch & Sermanet, 2020) dataset consists of 160k simulated trajectories and 440k real trajectories where each trajectory contains a language instruction (e.g., "move blue cube to the right"), a sequence of visuomotor controls, and a sequence of image frames corresponding to the execution of the task. The original trajectories have short horizons (e.g., only moving one block).

**PALM-E VLM Policy.** We modify the original PALM-E 12B model (Driess et al., 2023) to condition on a goal image as additional input before decoding the text actions. The VLM is finetuned on either the original short horizon data or the long horizon simulated data using 64 TPUv3 chips for 1 day. The supervision for short-horizon baseline is the single step language instruction in the original data, whereas the supervision for long-horizon UniSim data is the scripted long-horizon language instructions chained together that generated the video data. Other model architecture and training details follow Driess et al. (2023).

**Simulated evaluation.** In setting up goal in the simulated environments, a subset of 3-4 blocks (randomly selected) are moved by 0.05, 0.1, or 0.2 along the x,y axes (randomly selected). The original observation space has $x \in [0.15, 0.6]$ and $y \in [-0.3048, 0.3048]$. So the modification of goal location corresponds to meaningful block movements. For executing the long-horizon VLM policy trained on UniSim data, we first sample one language instruction from the VLM, predict a video of 16 frames, and use a separately trained inverse dynamics model similar to Du et al. (2023a) to recover the low-level control actions, which we found to slightly outperform directly regressing on control actions from language outputs of the VLM. We execute 5 instructions in total, and measure the final distance to goal according to the ground truth simulator state. We 5 evaluations each with a different random seed for sampling the initial state and resetting the goal, and report the mean and standard error in Table 2.

## D.2 DETAILS OF RL POLICY TRAINING

**Stage 1 (Supervised Learning) Model Architecture** The PaLI 3B model trained on Language-Table uses a Vision Transformer architecture G/14 (Zhai et al., 2022) to process images, and the encoder-decoder architecture of UL2 language model (Tay et al., 2022) for encoding task descriptions and decoding tokens which can represent language, control actions, or other values of interest (described below). **Objectives** In the first stage of training, using a dataset of demonstrations, we finetune the pretrained PaLI 3B vision language model checkpoint (Chen et al., 2022b) with the following tasks:

- **Behavioral Cloning:** Given observations and task instruction, predict the demonstration action. The continuous actions of the Language-Table domain are discretized into the form "+1 -5", and represented using extra tokens from the PaLI model's token vocabulary. As an example, "+1 -5" is represented by the token sequence (<extra_id_65>, <extra_id_1>, <extra_id_66>, <extra_id_5>).
- **Timestep to Success Prediction:** Given observations and task instruction, predict how many timesteps are left until the end of episode (i.e. success). Similar to actions, the number of steps remaining is represented via extra tokens from the PaLI model's token vocabulary.
- **Instruction Prediction:** Given the first and last frame of an episode, predict the task instruction associated with that episode.

We use learning rate 0.001, dropout rate 0.1, and batch size 128 to finetune the PaLI 3B model for 300k gradient steps with 1k warmup steps on both the simulated and real Language Table dataset similar to RT-2 Brohan et al. (2023).

**Stage 2 (RL Training) Reward Definition** As mentioned above, during Stage 1, given an observation and goal, the PaLI model is finetuned to predict how many timesteps are left until the demonstration episode reaches a success state. Let us denote this function by $d(o, g)$. The reward we use during RL training is defined as $r(o_t, a_t, o_{t+1}, g) = -[d(o_{t+1}, g) - d(o_t, g)] \cdot \mathcal{C}$, where $\mathcal{C} > 0$ is a small constant used to stabilize training ($\mathcal{C} = 5e - 2$ in this work). Intuitively, this reward tracks if from timestep $t$ to $t + 1$ the policy arrived closer to accomplishing the desired goal. Before starting Stage 2, we make a copy of the Stage 1 model checkpoint and keep it frozen to use as the reward model for RL training. **Environment Definition** To implement video generation as environment transitions, we expose the inference interface of the video generation model

through remote procedure call, and use the DeepMind RL Environment API (also known as DM Env API) (Tassa et al., 2018) to wrap the remote procedure call in the step function of the environment. When the environment is reset to start a new episode, a goal instruction is randomly sampled from the ones available in the dataset of demonstrations used in Stage 1. **RL Method** We initialize the RL trained policy using the Stage 1 checkpoint, which as mentioned was also trained with a Behavioral Cloning objective. A collection of actor processes perform policy rollouts in the video generation environment, and add rewards to the trajectories using the reward model defined above. The policy is updated using the REINFORCE (Williams, 1992) objective, i.e. $\nabla_\pi \mathcal{L}(o_t, a_t, g) = \nabla_\pi \log \pi(a_t|o_t, g) \cdot \left[ \sum_{i=t}^{T} \gamma^{i-t} \cdot r(o_i, a_i, o_{i+1}, g) \right]$, where $\mathcal{L}(o_t, a_t, g)$ represents the loss associated with the observation-action pair $(o_t, a_t)$ in an episode with the goal $g$. The actors are rate limited to prevent generated trajectories from being very off-policy. We report the hyperparameters associated with RL training in Table 7.

| Hyperparameter | Value |
|---|---|
| Max steps per episode | 100 |
| Number of actor processes | 64 |
| Number of image history stack | 2 |
| Learner batch size | 64 |
| Discounting factor $\gamma$ | 0.9 |

Table 7: Hyperparameters for training the VLA RL policy using the ACME framework.

### D.3 DETAILS OF VIDEO CAPTIONING

Note that even though UniSim is a video based simulator trained to condition on past history, we can achieve text-only conditioning by inputting placeholder frames such as white images while increasing the classifier-free guidance strength on text. We found this to work well in generating videos purely from captions of ActivityNet Captions. For generating data to train VLMs, we take the training split of ActivityNet Captions which consists of 30,740 text-video examples after the 50/25/25% train/val1/val2 split as in Chen et al. (2023). For each of the 30,740 text, we generate 4 videos from UniSim, and use the text labels as supervision in finetuning PaLI-X. As a result, we have 4X amount of the original training data (in terms the number of videos). In addition, we found the generated videos to generally align better semantically than the original ActivityNet Captions videos, which could contain noise and ambiguous videos that could be labeled differently. We use ground truth temporal proposals at evaluation following Chen et al. (2023) and Krishna et al. (2017). Following Chen et al. (2023) and Wang et al. (2021), we use the val1 split for validation and val2 split for testing.

## E ADDITIONAL ABLATIONS

### E.1 ABLATIONS OF DATASETS

We conduct ablations on dataset used in UniSim by computing the FVD and CLIP scores over 1024 samples from the test split. We observe that including internet data and various activity and robot data performs the best. Removing the internet data led to significantly worse FVD, highlighting the importance of using internet data in UniSim.

| Dataset | FVD ↓ | CLIP ↑ |
|---|---|---|
| Internet only | 219.62 | 22.27 |
| Without internet | 307.80 | 21.99 |
| Universal simulator | **211.30** | **22.63** |

Table 8: **Ablations of datasets** using FVD and CLIP score on the held-out test split. Including internet data and diverse human activity and robot data in UniSim achieves the best FVD and CLIP scores.

### E.2 ABLATIONS OF MODEL SIZE

We conduct ablations on model size by computing the FVD and CLIP scores over 1024 samples from the test split. We found that while increasing the model size improves the video modeling performance, the amount of improvement measured by FVD plateaus as the model gets bigger, which is slightly disappointing from a scaling point of view.

| Model size | FVD ↓ | CLIP ↑ |
|---|---|---|
| 500M | 277.85 | 22.08 |
| 1.6B | 224.61 | 22.27 |
| 5.6B | **211.30** | **22.63** |

Table 9: **Ablations of model size** using FVD and CLIP score on the held-out test split. The largest model achieves the best FVD and CLIP scores.

# F    FAILED SIMULATIONS WITHOUT JOINT TRAINING

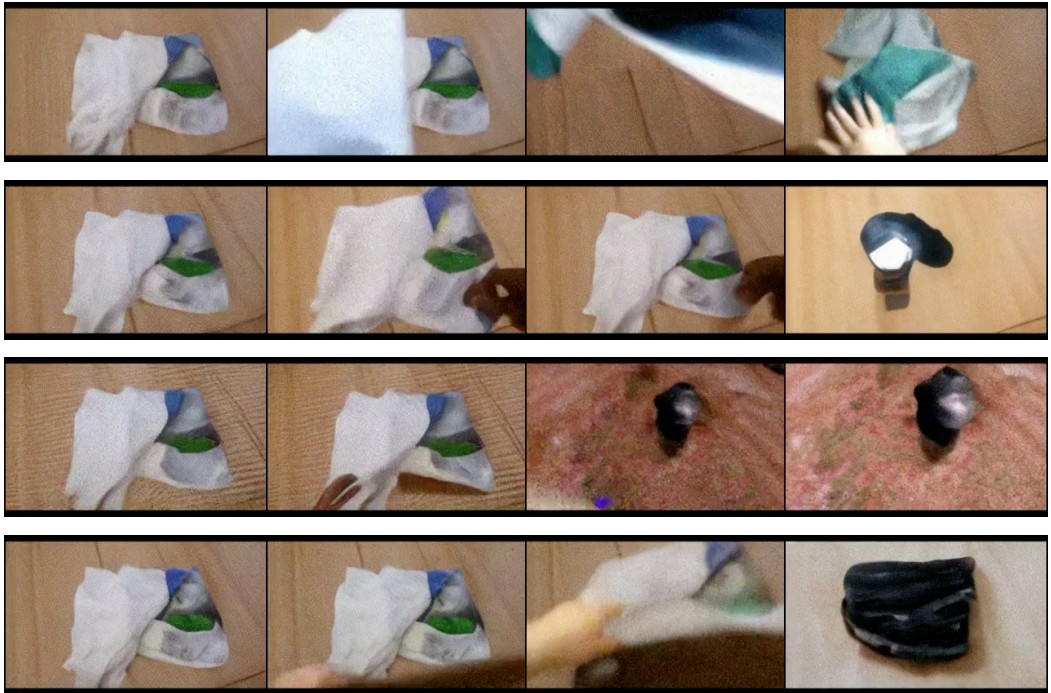

Figure 13: Failed environment simulation from the action "uncover bottle" without training on broad data as in UniSim. Top two videos are generated from only training on SSV2. Bottom two videos are generated from only training on generic internet data (without SSV2, EpicKitchen, Ego4D, and various robotics dataset).

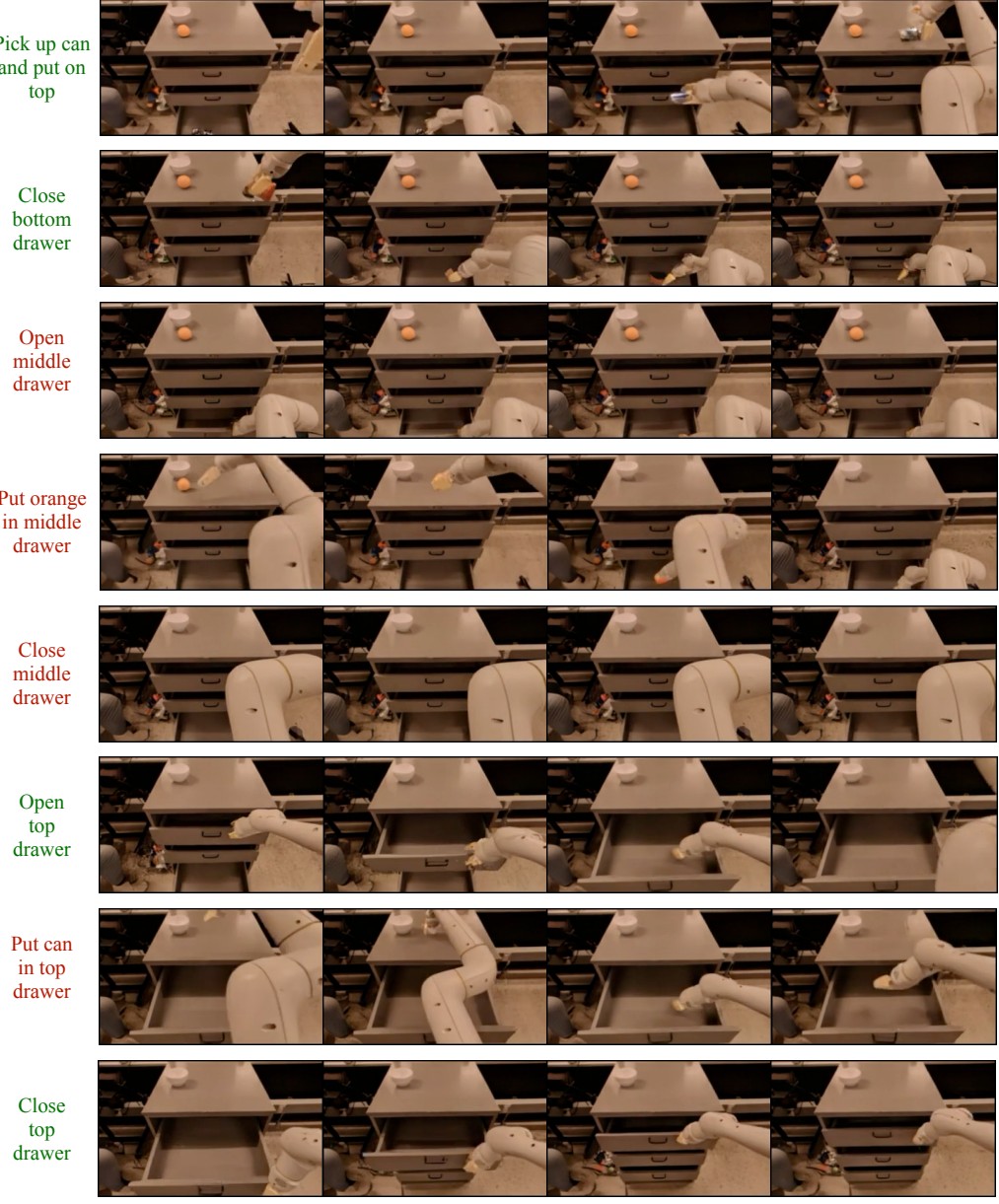

Figure 14: When the text-to-video model behind UniSim is only trained on data from Brohan et al. (2022) as opposed incorporating broad data from the internet and other manipulation datasets, long-horizon interaction simulations fail half of the time (red text).

