# OpenReview forum: "Learning Interactive Real-World Simulators"
_ICLR.cc/2024/Conference — ICLR 2024 oral_

### Official Review · Reviewer_ihUZ · 2023-10-31

**Soundness:** 3 good
**Presentation:** 3 good
**Contribution:** 4 excellent
**Rating:** 8
**Confidence:** 3

**Summary:**

This paper introduces a universal simulator (UniSim) that aims to simulate how humans and agents interact with the world. The proposed framework combines various types of datasets, including internet text-image pairs and robotics data, with the motivation that existing datasets are useful along different axes. The paper uses a video diffusion model as an interactive simulator of the world. UniSim can simulate both high-level instructions and low-level control, which show zero-shot transferability to real-world scenarios, addressing the sim-to-real transferability problem. The authors highlight the potential for UniSim to be used in broader applications, such as video captioning and rare event detection.

**Strengths:**

- This is an interesting paper that presents some exciting results.
- The paper is well organized and well-written.

**Weaknesses:**

- It would be nice if the paper delved more into the limitations of the models. The paper has shown that exciting results can be obtained, but it's useful for the community to know the limits of the generalization capabilities, especially if people want to use this in the future for various applications.
- For reproducibility, it would be helpful if the authors could release the code and some example pre-trained checkpoints.

**Questions:**

See weaknesses.

---

> ### Author Response · Authors · 2023-11-20
> **Author Response**
>
> Thank you for the positive feedback! We have expanded the discussion on the limitations of this work and included the following in the paper (conclusion section):
>
> - Hallucination. When an action is unrealistic given the scene (e.g., “wash hands” is given to a tabletop robot), we observe hallucinations (e.g., the table turns into a sink or the view turning away from the tabletop robot and a sink shows up). Ideally, we want UniSim to detect actions that are not possible to simulate as opposed to Hallucinate unrealistic outcomes.
> - Limited memory. UniSim conditioned on a few frames of the recent history cannot capture long-term memory (e.g., an apple in a drawer could disappear when the drawer is opened if putting the apple in the drawer is not a part of the history for conditioning). How much history to condition on depends on the application of UniSim (e.g., whether UniSim will be used for policy learning in a near-Markov setting or question answering that requires long-term memory).
> - Limited out-of-domain generalization. This is especially true for domains that are not represented in the training data. For instance, UniSim is mostly trained on 4 robot morphologies, and its ability to generalize to an unseen robot is limited. Further scaling up training data could help, as UniSim’s training data is nowhere near all the video data available on the internet.
> - Visual simulation only. UniSim is not suitable for environments where actions do not cause visual observation change (e.g., different forces in grasping a static cup). A true universal simulator should capture all aspects of the world beyond visual experience (e.g., sound, sensory, etc)
> We will work on releasing a code for doing inference with UniSim. We will try to open-source the UniSim model, but since the model is pretrained on private videos, we foresee challenges in this process.

---

### Official Review · Reviewer_b2Jg · 2023-10-31

**Soundness:** 3 good
**Presentation:** 3 good
**Contribution:** 4 excellent
**Rating:** 8
**Confidence:** 4

**Summary:**

This paper presents UniSim, a video prediction and generative model aiming for serving as a universal simulator of diverse scenarios conditioned on input language-described actions. It devotes a big effort in combining dataset with different modalities and information axes, trained a unified generative model, and shows the trained model can be used for downstream policy learning.

**Strengths:**

- Very cool and impressive research direction and proposed method
- Huge effort devoted in unifying multiple large scale datasets
- Experiments demonstrated effectiveness for downstream policy learning

**Weaknesses:**

I think the paper presents a very important step towards learning a universal video predictive world model. One of my questions is, the shown demo looks like generally still in distribution, in terms of generalization across different embodiment: the generated video contaiing robot are very similar to robotic dataset, and in more complex scenes training using human videos the model seems only handling human hands. How does it work in those complex scenes when the model is commanded to predict outcomes given a robot action input?
Also, when it comes to low level control input, the paper seems only handling delta motion in the cartesion space. Does it handle more general end-effector action in SE3 space? (joint space seems out of reach for this family of method since it's not observable) Is it true that for predicting outcomes conditioned on robot action, the robot arm needs to be visible in the first place?

**Questions:**

See above.

---

> ### Author Response · Authors · 2023-11-20
> **Author Response**
>
> Thank you for noting the significance of this work! We provide answers to your questions below.
>
> > Out-of-domain generalization for robotics.
>
> We note that because UniSim is only trained on 4 types of robots in fixed settings (e.g., in front of a cabinet, a tabletop with colored blocks), the out-of-domain generalization of UniSim is limited. We observe both success and failures (https://drive.google.com/file/d/1tpInX4KUywXLt971Tr15_RsW8I_ytpvu/view?usp=sharing) applying UniSim to out-of-domain settings. We believe that training UniSim on more diverse robotic datasets (e.g., RT-X [1]) can potentially mitigate this limitation. We made note of this limitation in the conclusion section of the paper.
>
> > More general end-effector action in SE3 space.
>
> We finetuned the UniSim model on the RT-X dataset [1] using low-level control actions (e.g., 7 DoF end-effector). We provide examples (https://drive.google.com/file/d/1AiMFfCEPIL2GXIYXu2qDThvtXzV2I4-o/view?usp=sharing) where UniSim can visually simulate the end-effector actions reasonably well. We note that these low-level control action conditioned models require more rigorous evaluation such as being used to train policies. We also hypothesize that UniSim can potentially be effective as long as there is an action interface shared by robots from which abundant data can be combined. Texts, cartesian coordinates, and end-effectors are all examples of such shared action spaces.
>
> > Conditioned on robot action, the robot arm needs to be visible in the first place?
>
> The robot arm does not need to be visible to simulate, for instance, the ego-centric view from a robot's wrist view, which is quite similar to ego-centric videos with camera settings converted to actions. Here is an example (https://drive.google.com/file/d/1EBqTpONGMyF9ZZCHA9kEAxnR_Yu9ewGY/view?usp=sharing) simulation of a robot’s wrist view while stacking blocks.
>
> [1] Open X-Embodiment: Robotic Learning Datasets and RT-X Models.

---

> > ### Comment · Reviewer_b2Jg · 2023-11-22
> > **Reviewer Response**
> >
> > Thank you for your reply! I will keep my positive score.

---

### Official Review · Reviewer_kkjZ · 2023-11-02

**Soundness:** 3 good
**Presentation:** 4 excellent
**Contribution:** 3 good
**Rating:** 8
**Confidence:** 5

**Summary:**

- The paper presents a video diffusion model that does conditional next (few) frame prediction. It conditions on previous frames and either a text description of the video, or more granular actions (robotic movements, or camera angles). The focus is on its use in robotics contexts.
- The novelty is in the mix of data trained on. Rather than focusing on a single environment or even single action space, the model (UniSim) is trained jointly on 14 common datasets, from the text-image LAION dataset (often used for image generation), to the Something-somethingV2 video dataset (often used for video classification). Significant compute is used (512x TPUs)
- A limited ablation is conducted on how previous observations should be conditioned upon.
- Three use cases are explored:
1) A separate vision-language model is first trained to predict language instruction and actions, given a start and end observation, on a robotics control task in the Language Table environment. It is then finetuned using simulated trajectories from data synthetically generated by UniSim (longer to those in the original dataset).
2) A separate policy model is first trained via BC on the Language Table environment, then finetuned with RL using simulated trajectories from data synthetically generated by UniSim (itself trained on Language Table data).
3) A separate video-to-caption model is found to benefit when finetuned on data synthetically generated by UniSim, for producing captions on ActivityNet.

___
Following the rebuttal, I upgrade my ratings as follows: soundness 2$\to$3, overall rating 5$\to$8, and confidence 4$\to$5. The main remaining weakness is that the mixture of datasets and modalities (a key contribution of the work) appears to be of limited benefit on the tasks assessed by the paper. But there are enough positives in the paper for me to downweight this issue.

**Strengths:**

- The paper will undoubtedly draw a lot of attention and excitement from researchers working in several areas, including RL, robotics, large models, and diffusion models.
- It represents a major effort in training a cross-domain model, with emphasis on its use for robotic control.
- I welcome this kind of larger scale-up work being submitted to an academic conference, since a recent trend has seen similar works restricted to industry-lab preprints.
- The first few pages motivating the work are quite inspiring.
- Effort has been made to explore a range of use cases.
- Overall it represents a very promising direction towards foundational models for control.

**Weaknesses:**

I expect that this paper will comfortably clear the bar for acceptance. However, there are two main issues I believe should first be addressed. I've set my score relatively low because of these, but anticipate increasing it following a revised version.

1) Whilst it's difficult to accuse the paper of overclaiming in any specific place, the writing and framing risk feeling a little showy. The title is very general, and applies to any paper on model-based RL for the real-world rather than something specific to this paper, and naming the method a "universal simulator" feels grandiose. (Happy to collect other revewiers' opinions on this.) The connection between POMDP's and action-conditioned video generation is more-or-less assumed by any world model paper (e.g. [1]), and shouldn't be highlighted as a main contribution of the paper.
2) One of the recurring claims throughout the paper is that the major novelty is UniSim's "orchestration of diverse datasets, each providing a different aspect of the overall experience" into a single model. Yet no hard evidence is given for this combination being important -- aside from two vague figures in Appendix E. At a minimum, it would be important to train a version of UniSim on say, datasets from the Language Table environment _only_, and report numbers for when synthetic data was generated from this, in Table 2 and 3. This would help support the claim that dataset diversity is valuable.

Other issues (in decreasing priority)
- I think it'd be useful to investigate how entwined the effect of actions is with the dataset distribution. For example, could camera commands (zoom in etc) successfully be applied to kitchen scenes as in Figure 3? The fact that the name of the dataset had to be included as part of the action during training, makes me suspect actions may not be able to generalise well to new kinds of video. This would not be a dealbreaker for the paper's acceptance, but is important to show readers how general this data mixing is.
- A lack of strong baselines might be expected for this kind of scale-up work. But in their absence, ablations become more important, to verify that the various components of the model were all necessary. The paper only presents a brief study of which frames to condition on.
- The model section is poorly written. The use of $\mathcal{T}$ is (I think) slightly misleading -- usually the transition fn of a POMDP is defined as operating on the states, $\mathcal{T}(s_t,a_t) \to s_{t+1}$, and there is a separate emission function producing the observations, $p(o_t|s_t)$. Eq. 1 implicitly combines these -- I might recommend renaming it $f$ or $g$ or whatever. I didn't follow why $o_l$ notation needed to be introduced, since it's immediately unrolled into $[o_t, o_{t+1}]$ and never referred to again. I also didn't understand why the model conditions on the noised, rather than clean, previous observations. It's said the last four frames are concatenated from $o_{t-1}$, which confused me -- does $o_{t-1}$ represent four frames, or should it read $o_{t-1:t-4}$ or similar?
- It's a shame to give the model details only in the Appendix C, as I believe many readers would be interested in them. I hope some of these can be shifted to the main body, particularly key details around the diffusion architecture (such as the core and super-resolution modules) and the amount of compute required.
- Any algorithmic or model novelty is light (more or less straightforward video diffusion).
- The two main experiments were conducted on environments that were within the training distribution of UniSim. It would have been more impressive to investigate the performance on new environments.
- The wordy description of all datasets in 2.1, I felt was much better summarized by Table 5 in the Appendix (perhaps with the addition of a column explaining how an action space is defined and handled), and might be swapped. (Optional!)

Minor issues/questions
- Appendix says 1M steps on 512 TPUs with batchsize 256 -- this seemed a low ratio of training updates to available compute. Did performance saturate beyond this?
- What was the wall clock time of the model training?
- How many parameters were in the model?
- Will the model weights be open-sourced?

[1] Transformers are Sample-Efficient World Models

**Questions:**

See weaknesses.

---

> ### Author Response · Authors · 2023-11-20
> **Author Response**
>
> Thank you for the detailed and constructive feedback! We answer your questions below and in the updated paper. Please let us know if you have additional questions.
>
> > 1.1 Novelty in relation to prior model-based RL work.
>
> While UniSim shares similarities with model based RL, model-based RL generally focuses on single-task settings and often do not use rollouts in the pixel space for policy optimization. UniSim focuses on learning a general purpose simulator that can be used for a variety of purposes such as serving as environments for manipulation and navigation, supporting human-AI interaction, generating data for VLMs, etc. More importantly, existing model-based methods have yet to achieve good real-world simulation results in the pixel space, so it is hard to call them “Interactive Real-World Simulators” as in our title.
>
> > 1.2 Concerns around method name.
>
> We were aware that our naming choice of 'universal simulator' could lead to concerns about overclaiming, similar to some other naming choices such as 'generalist agent.' However, we wanted to take a forward-looking stance when we chose this name, as UniSim demonstrates considerable potential in using internet data to simulate the universe, indicating that such an ambitious goal is achievable. We are open to renaming the method if reviewers generally agree that a less ambitious name is more favorable.
>
> > 1.3 Novelty in establishing the connection between video generation and POMDPs.
>
> Thank you for noting that Micheli et al. have previously established this connection. We have reformulated UniSim as an observation prediction model that approximates sampling in a POMDP. This formulation allows us to use multi-frame history conditioning to produce consistent long-horizon videos. We have also updated the second contribution bullet in the introduction.
>
> > 2. Effect of diverse datasets.
>
> Thank you for pointing out the lack of dataset analysis in the current version of the paper. We conducted ablation studies on the effect of diverse datasets, and included the following table in Appendix E.1:
> | Dataset | FVD | CLIP |
> | ----------- | ----------- | ----------- |
> | Internet only | 219.62 | 22.27 |
> | UniSim without internet | 307.80 | 21.99 |
> | UniSim   |   **211.30**     | **22.63** |
>
> We found combining internet data with various human activity and robot data results in the best FVD and CLIP scores on a held-out test split, whereas only using internet data (last row in Table 5) or not including internet data (all except for last row in Table 5) leads to worse performance. We further provide more qualitative examples (https://drive.google.com/file/d/174qUvW8fUxvKBVv3UV4m2yZ8y3v6gMW6/view?usp=sharing) where training on entire UniSim data is better than using internet-only data or without internet data. In terms of using downstream RL and planning to evaluate the effect of diverse data, we trained a video model only on Language Table as suggested by the reviewer, but we did not observe significant performance differences between UniSim and this task-specific video model in Table 2 and Table 3. Our hypothesis is that Language Table is a simple domain and already has a lot of data, so training a domain-specific video model is sufficient, but this is not true in general for harder environments.
>
> > 3.1 Entwined actions and videos.
>
> We found that actions are generally not entwined with videos in the training distribution; we can apply diverse actions to scenes as long as such actions are reasonable. As the reviewer requested, we generate various egocentric movements in the kitchen scene (https://drive.google.com/file/d/1HhvJIOlA2TAotZiRrt0CHVkgjc0kEQH2/view?usp=sharing) from Figure 3, including look up, look down, zoom in, zoom out, etc. We see that the generated videos generally follow the actions, but the videos might look less realistic when the actions are less reasonable (e.g., zooming out from a person’s view looking down at the kitchen table). We also generated some diverse interactions with objects (https://drive.google.com/file/d/1MWzWOHpeC9etJsW6gZXHm2q2E8DKyR24/view?usp=sharing).
>
> > 3.2 Dataset name as a part of action label.
>
> Note that we only used dataset name for dataset with very few videos (habitat), as it helped the model identify the domain better. We think this may not be needed if we tune the data mixture ratio during pretraining. For all the other datasets, we don’t use dataset name, so that language actions from different datasets can generalize across domains.

---

> > ### Author Response · Authors · 2023-11-20
> > **Author Response**
> >
> > > 4. Additional ablations.
> >
> > Thank you for pointing out this weakness of the paper. In addition to adding the ablation of dataset diversity in Appendix E.1 and additional qualitative examples, we also conducted an ablation on the model size in Appendix E.2:
> >
> > | Model size | FVD | CLIP |
> > | ----------- | ----------- | ----------- |
> > | 500M | 277.85 | 22.08 |
> > | 1.6B | 224.61 | 22.27 |
> > | 5.6B   |   **211.30**     | **22.63** |
> >
> > We found that while increasing the model size improves the video modeling performance, the amount of improvement measured by Fréchet Video Distance (FVD) plateaus as the model gets bigger, which is slightly disappointing from a scaling point of view.
> >
> > > 5.1 Confusion around the POMDP formulation.
> >
> > Thank you for pointing out the confusion. We have reformulated UniSim as an observation prediction model $p(o_t | h_{t-1}, a_{t-1})$ in Section 2.2 (Page 4), where $h_{t-1}$ denoting the history up to time $t-1$ corresponds to the belief state in a POMDP. Under this formulation, sampling from $p(o_t | h_{t-1}, a_{t-1})$ corresponds to doing rollouts in the POMDP. We also provided motivation for this formulation (i.e., directly learning the POMDP requires learning distributions over $s_t$, which is difficult). We also clarified that in the POMDP of the real world, the frequency and duration of interactions vary (e.g., a human opens a door in 2 seconds, a motor executing a robot control in 0.2 second), so $o_t$ is a set of (variable length) frames and $a_t$ is a set of (variable length) control sequences or some high-level text actions. For data from most environments, we use the last four frames of $o_{t-1}$ for conditional generation of $o_t$. However, when $o_t$ only consists of a single frame, we condition on $o_{t-4},...,o_{t-1}$ to generate $o_t$, as both the last 4 frames of $o_{t-1}$ and $o_{t-4},...,o_{t-1}$ are specific instantiations of $h_{t-1}$.
> >
> > > 5.2 Why the model conditions on the noised, rather than clean, previous observations.
> >
> > The model is conditioned on the *clean* previous observations concatenated with the noise samples for the next set of observations. We made the wrong choice to call this concatenation the noisy previous observation, which has now been corrected.
> >
> > > 6. Model details in main body as opposed to Appendix C.
> >
> > We added a paragraph summarizing the architecture and training on Page 5 and point the rest of the architecture and training details to the appendix.
> >
> > > 7. Limited algorithmic novelty.
> >
> > The creation of UniSim did involve innovations around how to leverage diverse datasets, how to condition on past information in diffusion models, and how to handle actions at different levels of  abstraction. We think many algorithmic aspects of generative models are already there; a lot of room for innovations lies in how to scale up (e.g., combing datasets) and properly use generative models for tasks (e.g., through RL, planning, and generating training data).
> >
> > > 8. Out-of-domain performance.
> >
> > UniSim exhibits limited out-of-domain performance with both success and failure cases (https://drive.google.com/file/d/1tpInX4KUywXLt971Tr15_RsW8I_ytpvu/view?usp=sharing). We believe that training UniSim on more diverse robotic datasets (e.g., RT-X [1]) and scale up the data to more internet videos can potentially mitigate this limitation. We made note of this limitation in the conclusion section of the paper.
> >
> > > 9. Wordy dataset description.
> >
> > We made that section more concise highlighting different information covered in each dataset (the main point of that section). Thanks for the suggestion for including the dataset table in the main text, we will take this into account in final version.
> >
> > > 10. 512 TPUs with batch size 256.
> >
> > The model is large and we needed model parallelism, and hence the small batch size. The video modeling metrics converged with 1M steps.
> >
> > > Clock time and parameter.
> >
> > UniSim has 5.6B parameters required 20 TPU days to train with 512 TPU-v3. We now report that in the paper (Page 5).
> >
> > > Open-source.
> >
> > We will try to open-source, but since the model is pretrained on private videos, we foresee challenges in this process.

---

> > > ### Author Response · Authors · 2023-11-21
> > > **Author Follow-up**
> > >
> > > Dear Reviewer,
> > >
> > > We wanted to follow up early to make sure that your concerns are being properly addressed. Please let us know if there are additional questions that we can provide further information / experiments on. Thank you.

---

> ### Comment · Reviewer_kkjZ · 2023-11-22
> **Rebuttal response**
>
> I appreciate the hardwork put into the rebuttal by the authors, both to my own and other reviewers' feedback, which has been managed excellently. I believe the paper is signficantly improved and will be increasing my score.
>
> I share several reactions to the rebuttal.
> - 1.1/1.2 -- Fair points. I will follow up with other reviewers in discussion about naming.
> - 1.3 -- Great. And POMDP section seems much improved.
> - 2 -- Thanks for adding the additional experiments, though they seem light on details for now (assume these will be added in a later version). E.g. what does internet only data comprise? what were the metrics computed over?
> - 2 -- It's disappointing that the Language Table-only model performed as well as full UniSim (did I read that correctly?). This is a critical point since the dataset mix is a main contribution of the paper. I take the authors' point that more benefit might be expected in harder tasks, but the onus is on the paper to demonstrate this concretely. Nevertheless there is enough creativity and positives in the paper that I couldn't recommend rejecting it on these grounds. I would however push for these results to be honestly reported in the main paper.
> - 3.1/3.2 -- Brilliant, appreciate the videos. Looks to generalize better than I'd expected.
> - 4 -- These seem good to me. I think it's a little harsh on the model to say the scaling behavior is disappointing -- the improvements are still significant for the 5.6B model, and there is no reason to expect these metrics would improve in the smooth power-law patterns seen in cross-entropy loss.
> - Remaining points look good.

---

> > ### Author Response · Authors · 2023-11-22
> > **Author Response**
> >
> > Thank you for the thoughtful response. We answer your questions below.
> >
> > - 1.2 For the method name, we wonder if it would help if we put less emphasis on "universal", and in fact explain that universal does not mean that the simulator can simulate everything, but that it uses a universal action-in, video-out interface, which could mitigate the risk of overclaiming.
> >
> > - 2. The internet data encompasses the text-image data from LAION-400M and ALIGN, and various miscellaneous videos from the internet that have not been previously published. The metrics were computed on a held-out set from Ego4D using 1024 samples. We will be sure to include these details in the final version of the paper.
> >
> > - 2. Yes, you read our response correctly. We will be sure to report that training a domain-specific simulator just for Language Table led to similar performance to UniSim in the final version of the paper. In addition to the reason listed in our initial response, such as Language Table is visually simple and already have abundant data (160k trajectories), Language Table is the only task that uses $\Delta x, \Delta y$ as low-level control actions in the data mixture of UniSim. We expect benefit from joint training when there are more datasets that share this Cartesian action space, but fewer trajectories in each of the datasets.
> >
> > - 4. Thank you for pointing out the difference in the scaling behavior between these video modeling metrics and the scaling behavior of large language models. We agree that there is still significant effect to scaling up UniSim, and it would be interesting to further scale up UniSim to even larger models and more datasets.

---

### Official Review · Reviewer_ZLpe · 2023-11-03

**Soundness:** 3 good
**Presentation:** 3 good
**Contribution:** 2 fair
**Rating:** 6
**Confidence:** 4

**Summary:**

In this work, the authors propose to learn a universal simulator (UniSim) of real-world interaction through generative modeling (a diffusion model for outputting the next frame given the previous frame and the input actions). They achieve so by careful orchestration of diverse datasets, which are rich along completely different axes (e.g., some videos have object-level diversity, some have densely labeled language instructions, and some have scene-level diversity). They show applications of the proposed simulator such as training long-horizon embodied planners and low-level object manipulators.

**Strengths:**

+ Reasonably scalable approach to collect training data for the proposed simulator
+ The use of diffusion models to fuse different aspects of the diverse datasets with decent results is impressive
+ Particularly the sim-to-real transfer is a promising direction for using the proposed real-world simulator.

**Weaknesses:**

While this work shows great promise in a range of downstream applications. I believe it might need more experimental evidence to support the claim that it can simulate low-level actions well. Specifically, section 4.2 only shows results for a relatively simple object (mostly blocks) re-arrangement (without grasping, e.g.) on a table. What about grasping objects, pulling objects (e.g., opening a drawer), etc? It will give us insights as to how fine-grained the controls are supported by the proposed simulator, even if it cannot simulate low-level actions perfectly.

**Questions:**

See “weaknesses”

---

> ### Author Response · Authors · 2023-11-20
> **Author Response**
>
> Thank you for the feedback. We finetuned the UniSim model on the RT-X dataset [1] using low-level control actions (e.g., 7 DoF end-effector). We provide examples (https://drive.google.com/file/d/1AiMFfCEPIL2GXIYXu2qDThvtXzV2I4-o/view?usp=sharing) where UniSim can visually simulate the end-effector actions reasonably well. We note that these low-level control action conditioned models require more rigorous evaluation such as being used to train policies. We also hypothesize that UniSim can potentially be effective as long as there is an action interface shared by robots from which abundant data can be combined. Texts, cartesian coordinates, and end-effectors are all examples of such shared action spaces.
>
> [1] Open X-Embodiment: Robotic Learning Datasets and RT-X Models

---

> > ### Author Response · Authors · 2023-11-21
> > **Author Follow-up**
> >
> > Dear Reviewer,
> >
> > In addition to the low-level action simulation above, we also provide simulations on another 7 DoF robot interacting with a kitchen cabinet using low-level robot actions as input: https://drive.google.com/file/d/1eM2i6VtR93zZ5JcWCHG2EpYCqtjPioH8/view?usp=drive_link. We input a sequence of low-level actions both as text strings and as discretized control values as described in Section 2.1 of the paper.
> >
> > We would like to ask if your concerns regarding more complex simulation conditioned on low-level control actions addressed, and if there is anything preventing you from increasing your score. Please let us know, and thank you for your time.

---

### Author Response · Authors · 2023-11-20
**Author Response Summary**

We thank all reviewers for the feedback. We conducted additional experiments and incorporated the results in the paper:
- In response to Reviewer ZLpe and b2Jg’s question about low-level robot action simulation,  we trained UniSim condition on low-level action controls from RT-X [1]. We observed that UniSim can indeed condition on low-level end-effector actions to generate reasonable simulation results.
- In response to Reviewer kkjZ’s question about the effect of diverse datasets and additional analysis, we conducted additional ablations on the effect of training data and model size. We reported quantitative metrics highlighting the importance of including diverse data in training.
- In response to Reviewer kkjZ and b2Jg’s question about out-of-domain generalization, We conducted experiments to investigate this, and provided both success and failure cases that we have observed.

In terms of major writing updates to the manuscript (highlighted in blue in the updated manuscript):
- In response to Reviewer kkjZ, we updated Section 2 by reformulating the connection between UniSim and POMDPs through an observation prediction model.
- In response to Reviewer ihUZ, we included detailed discussion on the limitations of UniSim in the conclusion section.
Please let us know if you have any additional questions.

[1] Open X-Embodiment: Robotic Learning Datasets and RT-X Models.

---

### Meta-Review · Area_Chair_zK5v · 2023-12-03

**Metareview:**

The paper has received positive reviews from the reviewers (8, 8, 8, 6). It proposes a very elegant idea of simulation using real-world data. Training in the real world is challenging due to the time-consuming nature of the process. Consequently, simulation is preferable. However, simulation using synthetic data is not desirable since it may lead to poor generalization due to the domain gap. This paper combines the best of both worlds and proposes a simulator that generates real data. The reviewers raised some concerns such as novelty in relation to prior model-based RL work, the need for more general end-effector actions in SE3 space, and being a visual only simulator, but the rebuttal addresses those concerns effectively. The AC strongly supports accepting this paper as it has the potential to open up new horizons in the domain of Embodied AI.

**Justification For Why Not Higher Score:**

N/A

**Justification For Why Not Lower Score:**

This paper has proposed an elegant technique with promising results. It will be of interest to a broad audience, so it is recommended to be accepted as an oral presentation.

---

### Decision · Program_Chairs · 2024-01-16

Accept (oral)